# Understanding the inclusion and participation of adults from Black African Diaspora Communities (BAFDC) in health and care research in the UK: a realist review protocol

Eleanor Hoverd ![ORCID],[1] Violet Effiom,[2] Dionne Gravesande,[3] Lorna Hollowood,[4] Tony Kelly,[5] Esther Mukuka,[6] Taiwo Owatemi,[7] Ify Sargeant,[8] Shane Ward,[9] Rachel Spencer,[1] Dawn Edge,[10] Jeremy Dale ![ORCID],[1] Sophie Staniszewska[1]

For numbered affiliations see end of article.

**Correspondence to**
Mrs Eleanor Hoverd;
eleanor.hoverd@warwick.ac.uk

## ABSTRACT

**Introduction** People from Black African Diaspora Communities (BAFDC) experience poorer health outcomes, have many long-term conditions and are persistently under-represented in health and care research. There is limited focus on programmes, or interventions that support inclusion and participation of people from BAFDC in research. Through coproduction, this realist review seeks to provide a programme theory explaining what context and mechanisms may be required, to produce outcomes that facilitate inclusion and participation for people from BAFDC in health and care research, in the UK.

**Methods and analysis** A group of people from BAFDC with lived and professional experience, representing all levels of the health and care research system, will coproduce a realist review with a team of African-Caribbean, white British and white British of Polish origin health and care researchers. They will follow Pawson's five steps: (1) shaping the scope of the review; (2) searching for evidence; (3) document selection and appraisal; (4) data extraction and (5) data synthesis. The coproduction group will help to map the current landscape, identifying key issues that may inhibit or facilitate inclusion. Data will be extracted, analysed and synthesised following realist logic analysis, identifying and explaining how context and mechanisms are conceptualised in the literature and the types of contextual factors that exist and impact on inclusion and participation. Findings will be reported in accordance with Realist and Meta-narrative Evidence Synthesis Evolving Standards .

**Ethics and dissemination** The coproduction group will agree an ethical approach considering accountability, responsibility and power dynamics, by establishing a terms of reference, taking a reflexive approach and coproducing an ethical framework. Findings will be disseminated to BAFDC and the research community through arts-based methods, peer-reviewed publications and conference presentations, agreeing a coproduced strategy for dissemination. Ethical review is not required.

## STRENGTHS AND LIMITATIONS OF THIS STUDY

⇒ To our knowledge, this will be the first realist review undertaken exploring the contextual factors that influence inclusion and participation and identification of the mechanisms that facilitate inclusion and participation of people from Black African Diaspora Communities (BAFDC) in health and care research, internationally.

⇒ Coproducing the review with people from BAFDC with lived and professional experience, will facilitate a deeper understanding of current barriers and facilitators affecting the inclusion and participation of people from BAFDC in health and care research, acting as content experts, drawing on their experiential knowledge which is rarely reported in formal literature.

⇒ This study will help to inform the development of an intervention reflected in the methods that enable greater participation of people from BAFDC in health and care research in the UK and may also be applicable to other health and care research systems.

⇒ A limitation of the review is that the lead researcher (EH) is white British and does not have lived experience of being black. Thus, she will reflect on her positionality throughout with support of a diverse research team and patient public mentor who is from a BAFDC.

**PROSPERO registration number** CRD42024517124.

## INTRODUCTION

Research aims to enhance the health outcomes of the whole population.[1 2] However, the lack of participation (Participation in health and care research as defined by the UK's largest funder of research, the National Institute for Health and Care Research (NIHR), is when people take part in a research study.[3] This includes experiences of being recruited

into a study, such as a clinical trial, or participating in focus groups, or completing a survey, for example.[4] This differs from patient and public involvement engagement (PPIE) in health and care research, where patients and members of the public are actively involved as partners in the research process across a range of activities in, such as commenting on research materials, or being part of a steering group or contributing to the synthesis of results.[3]) in health and care research studies by underserved groups may contribute to these groups not reaping the benefits of scientific discoveries.[5–14] One of the most affected underserved groups is black and ethnic minority groups.[15]

Studies suggest that black and ethnic minority populations are most affected by health inequities and have historically been ignored, marginalised and forgotten in health and care research.[15–17] In the UK, Black African Diaspora Communities (BAFDC) are a minoritised group that includes black, black British, black Welsh, Caribbean or African, and individuals with dual/multiple heritage and other groups who have black African lineage.[18 19] Minoritised groups are defined as having their 'cultural, political and social power' destroyed because of their identity, by structures and processes that uphold power and domination.[19] For the purposes of this review, people from BAFDC refer to adults aged ≥18 years, as children and youths from BAFDC, warrant a separate review. Little progress has been made in improving health outcomes for individuals from BAFDC.[20–24]

In the UK, black women are four times more likely to die during pregnancy, or up to 1 year after giving birth, than white women,[25] type 2 diabetes prevalence is up to three times greater in BAFDC,[26] one in four black men will develop prostate cancer,[27] and the prevalence of hypertension and stroke are also higher in this population.[26 28 29] Such avoidable and unwarranted differences in health are the culmination of social, economic, environmental and structural disparities caused by the unequal distribution of power and resources, underpinned by structural inequalities and racism.[16 30] However, it remains unclear as to how genetic determinants may contribute to the higher incidence of certain diseases among people from BAFDC, primarily due to a lack of representation of these populations in genomics research which may have potentially serious consequences on treatment decisions.[31 32] For example, doses for the drug warfarin, a blood-thinning drug, have been reported as being prescribed based on an individual's race and ethnicity, despite genetic studies mainly being conducted on European and white Americans.[31] There is also evidence to suggest that there is some misunderstanding among biomedical researchers around the differences between sociopolitical constructs like race and ethnicity and biological factors such as genetic ancestry, which can lead to estimates of genetic heredity being biased if environmental and social determinants of health are taken into consideration during analysis.[33] Enabling representation of people from BAFDC in health and care research is critical for innovation and new discoveries; data science; ensuring access to specific therapeutic drugs, or treatments; building trust in science and medicine; addressing health disparities and ensuring provision of evidence-based healthcare.[14 34] Without this, there is a risk of perpetuating health inequalities and social injustice with data producing algorithms used to guide treatment decisions based on White populations.[34] Furthermore, the potential benefits of personalised medicine and access to new treatments such as in cancer trials may not be realised if people from BAFDC are excluded from research.[35]

The responsibility to improve participation has often been directed at individuals who are under-represented. Terms such as 'hard-to-reach' have been frequently used to describe individuals who are marginalised or disadvantaged, creating an unhelpful narrative that places the blame on individuals, as opposed to services being inaccessible due to a number of barriers such as studies not routinely reflecting the needs of underserved groups.[36 37] Health and care research is crucial for reducing health disparities and improving health and well-being globally, but without diverse participation and the representation of all groups, especially those who have some of the poorest health outcomes, social equity cannot be reached, creating an enduring challenge for some of the wealthiest, research-active countries.[2 14 23]

## The science of inclusion

In comparison to the UK, the USA has had a lengthy history of policies implemented to improve diversity within health and care research; consequently, the evidence base around inclusion and participation of black and ethnic minority populations is largely from the USA.[14 24 38–45] A report by The National Academies of Sciences Engineering and Medicine (2022) suggests that these policies have failed to reduce exclusion of underserved populations.[14] After the implementation of the National Institute of Health (NIH) Revitalisation Act of 1993, there was some reported increase in previously excluded groups, such as women, taking part in health and care research, though individuals from black and ethnic minority populations remained underrepresented.[44–47] Guidance around improving inclusion is problematic when attempting to operationalise, reproduce or measure it.[48 49] Adoption of NIH policies has not adequately encouraged researchers to widen their eligibility criteria, with the Federal Drugs Agency lacking the power to enforce recommendations, resulting in exclusion of the very groups the Act was developed.[14] Exclusion has been related to the lack of reporting of outcomes in research studies by race and ethnicity, impacting on the evidence base for clinical decision-making.[14] Exclusion is reported to have been observed in commercial cancer trials, where eligibility criteria can be very restrictive.[50–52] These trials often exclude individuals with comorbidities, which are more prevalent in black and ethnic minority populations, preventing access to new treatments, such as immunotherapy.[51 52] The impact of eligibility criteria

on inclusion requires further investigation to find solutions. Consequently, a move to develop a stronger body of knowledge about the science of inclusion is emerging, focusing on theory and empirical evidence related to the process of health and care research from initial engagement with individuals from black and ethnic minority populations to participation in studies.[14]

In the UK, the NIHR defines research inclusion as 'taking a whole systems approach to what we do and how we do it; identifying and removing longstanding, structural barriers to success across our people, policies, processes and practices'.[53] Robust evidence is needed to inform how the deficits in the infrastructure of health and care research systems should be addressed in order to focus on inclusion and participation.[54] There is a need to understand the contextual and causal factors that influence inclusion and participation and to develop the science of inclusion based on theory; clear explanations about theories of inclusion and participation of black and ethnic minority populations are lacking.[14 54]

### The importance of inclusion

In the UK, there has been less focus on policy measures around inclusion in health and care research (eg, there is a lack of practice and policy in regards to reporting of inclusion and assessments of quality of data collection), and more focus on recognising the importance of inclusion, and working to identify and address barriers to inclusion.[26 53] This contrast may reflect fundamental differences in the values and politics of the UK and USA and the long history of race and ethnicity data collection policies and standards in the USA.[48 55 56] The UK's NIHR has acknowledged the importance of inclusion in health and care research by developing and implementing a Research Inclusion Strategy 2022–2027 demonstrating commitment to a long-term inclusivity agenda.[53] A focus on widening access and participation for greater diversity and inclusion is a critical aspect of the strategy, indicating that 'work is required to initiate more inclusive recruitment strategies that promote participation from underserved groups nationwide'.[53] Creating a more equitable health and care research system is crucial to ensuring disadvantaged groups most affected by health inequities benefit from participating in research and feel that research is relevant to them.[26 39]

### Lack of progress

Globally, little progress has been made on participation of black populations in research studies.[14] There is strong evidence in the USA to suggest that African Americans are willing to participate in research, and the decision to participate may be influenced by mechanisms situated at other levels of the health and care research system (such as the intrapersonal and structural levels, rather than at an individual level).[14 49] There are reports that willingness to participate has been misrepresented due to biases held by researchers, though the mechanisms that cause bias to function at the intrapersonal and structural levels

of health and care systems are poorly understood.[14 49] It has been suggested that the responsibility for developing interventions that will improve inclusion and participation of people from BAFDC lies with researchers, institutions, decision-makers, policy-makers and funders of research.[14 57] Such interventions are required at multiple levels of the health and care research system including policy-making and funding to provide resources to a specific population with the aim of creating progressive change.[40] The way in which an intervention is developed and implemented is dependent on the power of individuals and institutions and how they manage its implementation, which is pertinent when considering the architecture required for an intervention, for a marginalised population such as BAFDC.[58] For example, if an intervention is designed and implemented by individuals from a perceived dominant population, it may lack the perspectives of a diverse population which may impact on the success of its design.[15] For an intervention to have the best chance of success, it is crucial to understand the context within which it will be implemented, as well as the mechanisms by which it works. Consequently, a realist approach provides the most suitable method for developing new thinking through offering a deeper, more pragmatic explanation than a systematic review.[59] While systematic reviews offer an assessment of the effectiveness of interventions, they often lack explanations as to how and why an intervention may, or may not have worked, which is particularly valuable in complex systems, such as the health and care research system.[59] The learning from this realist review will be a timely addition to existing efforts. We are at a pivotal point in addressing the structural, racial injustices that have been brought to the fore by critical events over recent years, such as the murder of George Floyd and the COVID-19 pandemic that disproportionately impacted black and ethnic minority communities.[16 60 61]

## METHODS AND ANALYSIS

Our positionality statement can be found in online supplemental file 1.

### Aims

This realist review will broadly explore secondary data to provide insights that will contribute to the development of a realist theory of inclusion and participation in health and care research by people from BAFDC in the UK. The theory may be applicable to other health and care research systems internationally. It will be undertaken with a coproduction group composed of people from BAFDC who have the lived experience of being Black as well as experience as patients, members of the public, healthcare professionals, community leaders, health and care researchers, research delivery staff, policy-makers and funders of health and care research. Thus, they are aware of the challenges related to inclusion and participation in health and care research for people from

BAFDC.[62] A realist review benefits from 'content experts' such as the coproduction group, improving the efficiency of the review through helping to focus the scope of the review, development of the initial programme theory and refining the search process and accurate interpretation of the results.[63] The review will help to explain the context of the health and care research system, identifying underlying causal factors that influence inclusion and participation of people from BAFDC in health and care research. We will develop a programme theory through following five key steps as recommended by Pawson *et al* that explains how inclusion and participation may be facilitated with people from BAFDC in health and care research.[64 65] This programme theory is intended to inform the development of an intervention to improve inclusion and ultimately outcomes for people from BAFDC.

### Review questions

1. What are the contextual and/or causal factors that influence inclusion and participation in health and care research by people from BAFDC?
2. How and where do mechanisms occur that underlie implicit and complicit bias that affects the inclusion and participation of people from BAFDC in health and care research?
3. What barriers and facilitators do people from BAFDC feel affect their experiences of participation in health and care research?
4. What might the components of an intervention(s) look like that foster inclusion and participation of people from BAFDC in health and care research?

### Study design

The design will be guided by Pawson *et al*'s five steps and in accordance with Realist and Meta-narrative Evidence Synthesis Evolving Standards for quality and reporting.[66] Figure 1 shows a diagram of the review process as adapted from Duddy and Wong.[67] Due to the complexity of context, the focus of the review will be steered by a unique theoretical framework developed with the coproduction group from pre-existing mid-range theories deemed pertinent to this research. Theories will be identified through analysis of resources from the review team, as well as comprehensive evaluation of the literature.[59] A realist review is a theory-driven method for investigating complex interventions or programmes.[65]

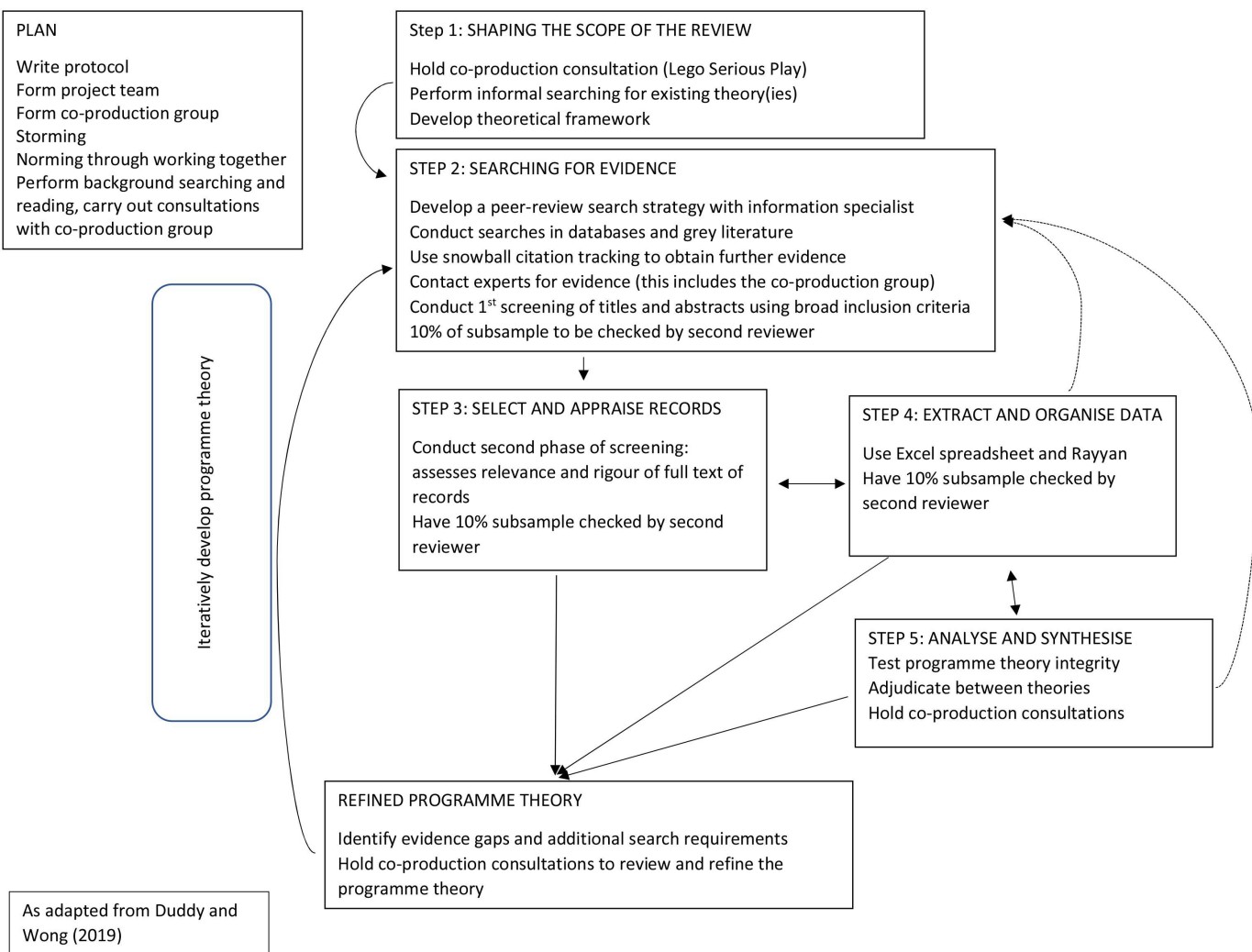

**Figure 1** Realist review project plan.

Evidence is produced in the form of theories that are best understood by analysing the contexts, mechanisms and outcomes of a programme.[59] Understanding context can help to create an optimal condition for the successful implementation of a programme or intervention.[68] The interaction between context (C) and mechanism (M) results in an outcome (O).[69] This formula is known as a CMO configuration (CMOC) and is a way of explaining theory in realist research.[59 69]

## Patient and public involvement

Stakeholder involvement is key to realist research and will be achieved by using a coproduction approach to PPI.[69] Stakeholders are people who may have an interest in the research being carried out, may be involved in shaping it and could be affected by the outcomes.[70] Coproduction is a way of sharing power, taking into account the skills and many perspectives within a group of people—it involves respecting and valuing the knowledge of others as well as providing reciprocity and focusing on building and maintaining relationships.[71]

### Recruitment

A coproduction group of eight individuals (VE, TK, DG, LH, EM, TO, IS and SW) from BAFDC will meet to share knowledge and identify training needs, around the realist approach, prior to the review. Individuals were approached by the lead author (EH) through her existing networks (some had already been involved in PPIE during the development phase of the research proposal): directly approaching black community organisations and individuals, providing written information about coproduction and the research project, with 1:1 meetings held virtually with all who were interested, to provide an opportunity to ask questions and meet EH. The coproduction group includes experts by experience, as well as stakeholders from the health and care research system who self-identify as belonging to BAFDC. In this review, experts by experience are regarded as patients and members of the public with lived experience of the health and care research system.[72]

### Work schedule

We will map out the review process and draw on expertise from within the group, through creative methods and discussion, to understand how the health and care research system currently serves people from BAFDC in the UK. The coproduction group will be consulted in alignment with the principles of coproduction throughout, as key 'experts' with additional stakeholders sought for expertise, if required.[70] The group will be consulted throughout the review via virtual meetings on Microsoft Teams and email, allowing for wider geographical inclusion as the group is based in the West Midlands and London regions, improving accessibility, with sessions kept to a maximum of 1.5 hours to avoid digital fatigue.[73] There will also be at least one face-to-face meeting, held at a central location convenient to the group, to allow for longer discussion and to help build relationships.

## Sequence of steps

The review will be conducted following five steps: (1) shaping the scope of the review, (2) searching for the evidence, (3) document selection and appraisal, which will be based on relevance and rigour, (4) data extraction and organisation of evidence will be conducted iteratively and (5) data synthesis will be used to explain deficits in context and how they impact on inclusion and participation and development of programme theory.

## Step 1: shaping the scope of the review: key concepts and construction of a theoretical framework

Programme theories create a scaffolding to explore how, why, for whom and under what circumstances complex programmes or interventions work.[74] In realist terms, an intervention is a theory.[74] This stage will focus on the development of a theoretical framework that will shape the focus of the review, support data extraction, appraisal and synthesis, and identify concepts around inclusion and participation of people from BAFDC in health and care research.[71] Evidence suggests that a small number of theories can be developed into a framework that will support the development of initial programme theory that incorporates the multiple levels of a system.[75] To identify relevant theories, concepts critical to inclusion and participation of people from BAFDC will be identified through creative discussion with the coproduction group using Lego Serious Play (LSP) as a method for facilitating their input.[76] A small pilot with NIHR Research Champions, or members of the public who have experience of being involved and engaged in health and care research will take place prior to using the approach with the coproduction group.

### Outputs

We will produce a simple, research participant pathway representing the health and care research process that will be used as an artefact, to consult with the coproduction group to stimulate discussion during LSP, to support the facilitation of individual's thoughts and reasoning (see figure 2).[77]

The coproduction group will use LSP to map the existing health and care research system. Concepts from other programmes in related areas can also be drawn on. Existing mid-range theories will then be identified through an initial scoping search based on key concepts that arise following the LSP activity. A list of these mid-range theories will be presented to the coproduction group to decide whether they should be included in the theoretical framework, based on their importance to the research aim and relevance to people from BAFDC. Theories deemed pertinent by the group will be merged into a framework to guide data extraction and appraisal. Evidence will be collected to theorise how an intervention may work, facilitating identification of CMOCs through

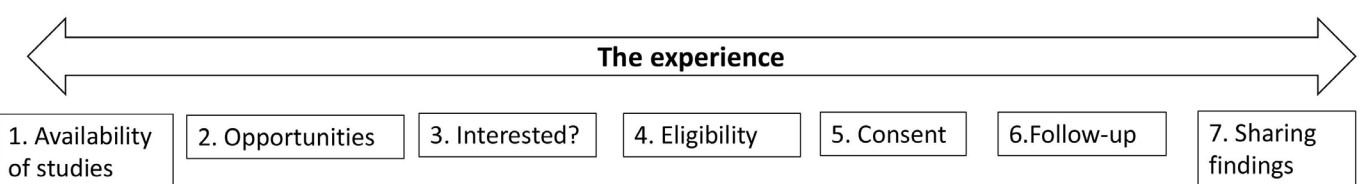

**Research participation pathway**

Pathway of research participation in health and care research for people from Black African Diaspora Communities (BAFDC)

| The experience |
|---|

| 1. Availability of studies | 2. Opportunities | 3. Interested? | 4. Eligibility | 5. Consent | 6.Follow-up | 7. Sharing findings |

**Figure 2** Pathway of experience of participation.

matching to existing theories, while considering information that may be missing from existing mid-range theories.[59 78]

### Step 2: searching for evidence

A search for evidence will be conducted, including international literature, collecting a wide range of secondary data sources and grey literature including: all types of studies, peer-reviewed articles, opinion pieces, commentaries, conference papers/proceedings, reports and social media excerpts.[62 65] In discussion with an information specialist, a pilot search of the evidence was conducted. The review will be conducted over a 6-month period including all literature that covers research that addresses the issues of inclusion and participation in health and care research in the period post World War II. There are no language restrictions. We expect further adjustments to the search strategy to be required. Discussions and decision-making with the coproduction group will be held around relevance of international literature to a UK context. A data extraction form will be informed by the theoretical framework as a way of combing the evidence and appraising it through scoring it as high, moderate, low based on its usefulness and relevance.[79] The data extraction form will include standard information including full reference, item type, country where source was published, quality and the group being researched, though will be developed iteratively to meet the needs of the review.[79]

### Data sources and search strategy

An initial search will be developed based on the theoretical framework, with key search terms identified in box 1. An example of the search strategy for Medline can be found in online supplemental file 2. EndNote will be used for storing all identified sources.[80] Search terms will be discussed with the research team and coproduction group and modified as required for each database or website.

Using the strategy, a search for the evidence will be conducted using the following databases: Medline, EMBASE, PsycINFO, Web of Science, Race Relations Abstracts, Sociological Abstracts, University of the West Indies and Patient-Centred Outcomes Research Institute. A search of grey literature will be conducted (box 2), although recommendations from the coproduction group may also evolve as the review progresses. Literature

sources will be prioritised by the coproduction group where relevant.

Once the initial search has been completed, we will decide if further iterative searching will be conducted which will be determined by the extent of literature identified and discussed with the research team and coproduction group. Search terms will be expanded in this instance with the information specialist.

Screening of abstracts will be carried out according to inclusion and exclusion criteria developed, based on the theoretical framework, with no date limit initially. This will be reviewed following the database search and a limit set, should the return of documents be excessive.

### Screening process

EH will undertake screening based on title and abstract and a second reviewer will screen a random sample of 10% of the identified citations, setting a kappa measure of k>0.8 to measure inter-rater reliability.[66] This will be calculated using the formula for Cohen's kappa.[81] Reviewers will discuss disagreements with the coproduction group and research team. If k<0.8 a further 10% sample will be screened, and the outcome discussed with the research team. Should an abstract not be available (eg, in some grey literature), documents will be included and full screening of each will be carried out at second level screening. Rayyan referencing software will be used for screening as it enables blinding of screening decisions, collaboration functions and presentation of screening

---

**Box 1    Search terms**

⇒ Health Services Research
⇒ Clinical Trials
⇒ Biomedical Research
⇒ Black
⇒ Afro-Caribbean
⇒ BME
⇒ BAME
⇒ African American
⇒ Black British
⇒ Participation, or Patient Participation, or Community Participation, or Stakeholder Participation, or Social Participation
⇒ Inclusion
⇒ Recruitment
⇒ Underrepresentation

---

**Box 2   Grey literature sources**

**Initial grey literature sources**
⇒ NHS Race and Health Observatory
⇒ Runnymede Trust
⇒ BRAP
⇒ The Health Foundation
⇒ The King's Fund
⇒ Operation Black Vote
⇒ The Voice newspaper
⇒ African Voice newspaper
⇒ West Bromwich African Caribbean Resource Centre
⇒ Enfield Caribbean Association
⇒ Caribbean and African Health Network
⇒ Vanderbilt University Medical Centre, Office of Health Equity Anti-racism hub
⇒ CARE (Community, Access, Recruitment and Engagement) Centre, Massachusetts General Hospital
⇒ Open Access Theses and Dissertations
⇒ Caribbean-studies@jiscmail.ac.uk
⇒ Caribbeanintelligence
⇒ Twitter, Facebook and YouTube
⇒ Coproduction group recommendations

---

**Box 3   Inclusion and exclusion criteria**

**Inclusion**
⇒ No initial date parameter.
⇒ All study designs (qualitative, mixed methods, quantitative, systematic reviews, etc) and grey literature.
⇒ All international sources will be included.
⇒ Document has relevance to development of programme theory, either the full text, or a section.
⇒ Population must be related to Black African Diaspora Communities (BAFDC), black British, African American, African or African-Caribbean people, though this may include sources that include information about other ethnic minority groups as well.
⇒ Must concern factors that affect inclusion and participation in health and care research with health and social care research.

**Exclusion**
⇒ Relates to BAFDC, black British, African American, African, African-Caribbean people <18 years old.

---

decisions in a simple format which can be exported onto a spreadsheet.[82] Papers that are excluded by one reviewer and included by another will be brought to the wider research team for further discussion, as well as those where one, or both reviewers are unsure.

### Iterative searching

As realist reviews are iterative, searching for additional data to support theory development is likely.[59] For each additional search, the research team will develop further inclusion/exclusion criteria, with each search tested and refined through support from an Information Specialist and screening conducted as summarised above.

### Step 3: selection and appraisal

After initial screening in step 2, documents will be selected based on relevance and rigour after the full text has been read by EH and inclusion/exclusion criteria applied[83] (see box 3). Relevant documents are those which are deemed helpful and valuable for building theory while rigour ascertains the trustworthiness and credibility of methods used to produce data.[84] Although, methodological quality of papers is not so important in realist research, relevance and robustness is.[85] Evidence may include a wide range of sources, for example, opinion pieces that may capture important evidence based on experiences.[83] Pawson asserts that even poor evidence, which may be methodologically weak, can provide rich insights and thus following a hierarchical approach to selecting documents should not pertain to realist reviews.[64 83 84] Realist research is informed by evidence, as opposed to evidence-based and therefore a diverse range of literature sources are needed to develop programme theories with sources selected according to relevance and rigour.[85]

### Step 4: extracting and organising data

Databases will be developed to extract and organise data, tailored for this review. Data will be extracted by EH with a 10% subsample of included papers, also completed by the second reviewer. Disagreements will be discussed with the research team and coproduction group. If kappa is <0.8 then a further subsample of 10% will be reviewed by the second reviewer. We will pilot the use of NVivo software as an alternative to Excel to assist data coding and organisation; we will code 3–5 documents in both databases to determine which is the most pragmatic tool for this review.[86] Extracted data will be centred on dialogue that helps to build programme theory.[87] Codes will be developed both inductively from documents, and deductively from the theoretical framework.[88] Data will be organised into broad, conceptual categories, through grouping the codes together and prioritising coding for causal insight in regard to the architecture of the health and care research system.[89] This will support identifying patterns in the data and configuring data into CMOCs.[88]

### Step 5: analysis and synthesis

Data analysis and synthesis will be led by EH with reflections and discussion among the research team and coproduction group to support interpretation of CMOCs using retroduction.[90] Retroduction allows the researcher to incorporate intuitions and insights, as well as inductive and deductive logic, considering the causal forces that may be producing what is observed.[90] This theorising is based on the researchers' interpretations and may not be testable in the first instance, particularly if only part of a theory is developed.[90] The coproduction group will be involved in coproducing this, to ensure that the developing programme theory, is true to their lived and professional experiences. The group will decide meeting frequency, location and length.

## Strengths and limitations

A key strength of this realist review lies with the coproduction group that will be involved throughout.[62] The combined skills and expertise of a group of stakeholders from BAFDC, with both lived and professional experience in the health and care research system will place people from BAFDC at the centre of this review. The group's lived experience of being black, or having a shared history of 'colonialism and enslavement in the past and continuing to experience racism and diminished opportunities' makes them experts in their own right and will ensure that their reality cannot be denied.[91] A limitation is that the lead researcher (EH) is white British and has not experienced racism. Therefore, she will reflect on her positionality throughout, with a willingness to engage, learn, and take on board the complex issues that may present. The coproduction group feel that Tuckman's model of 'forming, storming, norming and performing' is a useful pathway for reflecting on how the group may develop to ensure the aims and objectives of this project are met and to create a progressive group culture.[92] This review will be undertaken as a realist review, in order to answer timely research questions for which there is limited published evidence on interventions or programmes that effectively foster inclusion and participation in health and care research with people from BAFDC.[58 63] Limitations may include not being able to access or include all sources identified in searches as well as a limit on the number of iterative searches conducted due to timescales of the research. However, together as a research team and coproduction group, we will agree on the priorities of the programme theory with a view to conducting future research on additional aspects that we are unable to address in this project.

## ETHICS AND DISSEMINATION

This review does not require ethical approval due to the use of secondary data that already exists, for example, in databases and PPI in the form of coproduction. The review is registered with PROSPERO. However, ethical considerations around accountability, responsibility and power dynamics will be discussed and reflected on, with the coproduction group and research team. This review will lead to a programme theory that begins to explain what may work, for supporting inclusion and participation of people from BAFDC in health and care research internationally. This theory will continue to be built through qualitative data collection methods, with stakeholders from each level of the health and care research system, in a realist evaluation. Further outputs will include a set of principles, recommendations and a conceptual framework that will inform the development of an intervention suitable for implementation, to support a UK research system focused on equity and social justice.[14] An early prototype for an intervention, based on the findings, will be developed with the coproduction group. Outputs for people from BAFDC will be coproduced and will focus on

a series of community engagement events, dissemination of findings via radio, in newspapers such as The Voice, a well-established British African Caribbean newspaper since 1982 and via social media.

**Author affiliations**
[1]Warwick Medical School, University of Warwick, Coventry, UK
[2]NIHR Clinical Research Network West Midlands, Coventry, UK
[3]Public Contributor, Northampton, UK
[4]University of Birmingham, Birmingham, UK
[5]PPI Lead on Project, Birmingham, UK
[6]NIHR, London, UK
[7]UK Parliament, Coventry, UK
[8]Public Contributor, Stafford, UK
[9]Public Contributor, Sandwell, UK
[10]University of Manchester, Manchester, UK

**Contributors** EH conceived the study with input from SS, DE, JD, RS and TK. EH was responsible for the design and drafting of the protocol. SS, DE, JD, RS, TK, SW, EM, IS, VE, TO, LH and DG contributed to protocol development and SS, DE, JD and RS also provided methodological advice. SS, DE, JD, RS, TK, SW, EM, IS, VE, TO, LH and DG provided criticism and refinement of the manuscript. SS, DE, JD, RS, TK, SW, EM, IS, VE, TO, LH and DG approved the final version.

**Funding** EH is funded by the NIHR (Health Education England/National Institute for Health and Care Research Clinical Doctoral Research Fellowships programme, grant number 302121). SS is part funded by the NIHR Applied Research Collaboration (ARC) West Midlands (grant number NIHR200165)], the NIHR Health Protection Research Unit (HPRU) Gastrointestinal Infections (grant number NIHR200910), the NIHR HPRU Genomics and Enabling data (grant number NIHR200892), the NIHR West Midlands Evidence Synthesis Group (grant number NIHR153453) and the NIHR Health Determinants Research Collaboration Coventry (not yet awarded).

**Competing interests** None declared.

**Patient and public involvement** Patients and/or the public were involved in the design, or conduct, or reporting, or dissemination plans of this research. Refer to the Methods section for further details.

**Patient consent for publication** Not applicable.

**Provenance and peer review** Not commissioned; externally peer reviewed.

**ORCID iDs**
Eleanor Hoverd http://orcid.org/0000-0002-8482-655X
Jeremy Dale http://orcid.org/0000-0001-9256-3553

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
