## [Reviewer comments · BMJ Open]

ARTICLE DETAILS

TITLE (PROVISIONAL)	Understanding the inclusion and participation of adults from Black African Diaspora Communities (BAFDC) in health and care research in the UK: a realist review protocol.
AUTHORS	Hoverd, Eleanor; Effiom, Violet; Gravesande, Dionne; Hollowood, Lorna; Kelly, Tony; Mukuka, Esther; Owatemi, Taiwo; Sargeant, Ify; Ward, Shane; Spencer, Rachel; Edge, Dawn; Dale, Jeremy; Staniszewska, Sophie Editorial Board Member

VERSION 1 – REVIEW

REVIEWER	Bamford, Jordan The University of Manchester Division of Psychology and Mental Health
REVIEW RETURNED	15-Dec-2023

GENERAL COMMENTS	Peer Review Dear Editor, Thank you for the opportunity to peer review this article titled: 'Understanding the inclusion and participation of adults from Black African Diaspora Communities (BAFDC) in health and social care research in the UK: a rapid realist review protocol'. The paper I have reviewed presents a protocol for a rapid realist review, aiming to provide a programme theory that explains what context and mechanisms may be required to facilitate inclusion and participation of Black African Diaspora Communities in health and social care research in the UK. The protocol has an introduction, a methodology, a discussion of ethics and dissemination and reflection on the strengths and limitations of the proposed study. The introduction details evidence suggesting poorer outcomes for black people in the UK for an array of different health outcomes, and posits that to target health disparities, inclusive research is needed. The authors then present a detailed review of 'the science of inclusion', which details an in depth review of policies in the United States relating to inclusion in health and social care research – this at times feels irrelevant to the topic under investigation. The authors define inclusion, and describe the importance of inclusion and the lack of progress in inclusion. In the methodology section, the authors detail the review questions, and they detail how they will use a co-production group composed of people from BAFDC. The authors explain they will use a five step approach for the review. Shaping the scope of the review, searching for evidence, selecting and appraising, data extraction and data synthesis. They do not provide a comprehensive search strategy, or detail what information they will extract. They do detail what databases they will search.
---

	Authors then reflect on strengths and limitations of their review, and discuss ethics and dissemination of findings. They conclude on the potential of their review to help facilitate more inclusive research in the UK which could be helpful in an international context. This protocol relates to a very important topic, however I have concerns about the current draft of the protocol –  1) No comprehensive search strategy is provided (just search terms which are not exhaustive and some are counterintuitive, see major point 1). 2) This review is not registered with prospero currently. 3) The protocol is not referenced correctly, and many references in the introduction section are not related to the points the authors raise and need reviewed. 4) No data extraction form is provided, and as there is no limitation on study design it becomes very challenging to understand what the authors will be extracting from the studies included. I have focused my review into major, minor and other categories. Major  1. My main concern with the current protocol is a lack of a clear search strategy, and no explicit references to the proposed dates of the review ie what timeline do the authors propose for this search. On Page 34 the reader is presented with ‘search terms’. This is not a search strategy, and I believe usual convention would be to provide a clear example of the proposed search strategy that would be carried out, for example in Medline. I understand that the process of a realist review is iterative, however, an initial search will still be conducted and a protocol should detail that. Further, among the search terms, the terminology used by the authors themselves is not used, the BAFDC descriptor. I am also confused that the search terms include ‘African American’ when the review has a clear remit within the United Kingdom. Research which focuses on particular ethnic groups should be transparent with proposed terms that will be used in the search – the authors could inadvertently miss a particular population. 2. Page 22, line 513, states ‘the review will be registered with PROSPERO’ – this should be done before this article is considered for publication, so the reader has access and can review what has been registered by the authors. 3. Referencing needs reviewed. I believe the authors have referenced in Vancouver style, however from my reading, the references presented have not been numbered by the order in which they are cited and this is very confusing. For example on page 7, line 131 in the introduction there is references to [23,13, 27]. Yet this is the first time references 24 and 27 are presented, and therefore they should be 14 and 15. This needs corrected and the reference list updated consistently. The issue seems contained to the introduction section, but on correction this will change the entire ordering. 4. On page 18, lines 400-401 it states that ‘following a pilot search of the evidence it was felt that no time limit should be applied to the initial search’. If this is not a focused review on emerging evidence, why have the authors characterised their review as a ‘rapid’ review? 5. On pages 9 -10, the section ‘the science of inclusion’ is quite challenging to read/make sense of, and at times feels very unrelated to the topic of inclusivity in UK research. Authors may
--	---

	wish to reflect on whether this detailed analysis of US policy is needed. Minor 1. Page 7, lines 138-140. In the introduction it states 'Little progress has been made in improving health outcomes from individuals from BAFDC'. There is reference to MBBRACE-UK 2021 report which I believe is a great example of evidence for where little progress has been made to address ethnic differences in a particular health outcome. I am confused about the other references. The paper by Bello et al in Diabetologia, on quick review is a paper examining whether black West African men with type 2 diabetes present with greater hepatic and adipose tissue insulin sensitivity – for which there were no ethnic differences found. I do not understand how this provides evidence that there has been little progress made to improving health outcomes. I would recommend authors consider other evidence for poor outcomes among ethnic minority groups with respect to diabetic care, a good example on a quick review of the literature would be Mathur R, Farmer RE, Eastwood SV, Chaturvedi N, Douglas I, Smeeth L. Ethnic disparities in initiation and intensification of diabetes treatment in adults with type 2 diabetes in the UK, 1990–2017: A cohort study. PLoS medicine. 2020 May 15;17(5):e1003106. Further, references 19 and 20 are cited as evidence of limited progress, and these are links to the NIHR website. Reference 19 is to the NIHR glossary, under 'p'– on review I could not identify any empirical evidence for poor health outcomes for ethnic minority groups/the BAFDC in this section of the NIHR website, and further I have no indication as to what the authors are referring to. Reference 20 directs me to guidance from the NIHR on public involvement in research. There is a multitude of excellent references which could be used to assert the claim made by the authors, but the ones selected are in my opinion very unclear. References I would recommend and I believe could improve this section would be: Devonport TJ, Ward G, Morrissey H, Burt C, Harris J, Burt S, Patel R, Manning R, Paredes R, Nicholls W. A systematic review of inequalities in the mental health experiences of Black African, Black Caribbean and black-mixed UK populations: implications for action. Journal of Racial and Ethnic Health Disparities. 2022 Jun 29:1-3.. Mukadam N, Marston L, Lewis G, Mathur R, Rait G, Livingston G. Incidence, age at diagnosis and survival with dementia across ethnic groups in England: A longitudinal study using electronic health records. Alzheimer's & Dementia. 2023 Apr;19(4):1300-7.. Mathur R, Dreyer G, Yaqoob MM, Hull SA. Ethnic differences in the progression of chronic kidney disease and risk of death in a UK diabetic population: an observational cohort study. BMJ open. 2018 Mar 1;8(3):e020145. 2. On page 7, lines 141-142, it says 'black women are four times more likely to die during pregnancy...' and 'diabetes prevalence is up to three times greater in BAFDC' I assume this is in comparison to the white population in the UK, but I think this should be clarified for the reader. 3. Page 8, Line 155-158 states 'it remains unclear as to how genetic determinants may contribute to the higher incidence of certain diseases amongst people from BAFDC, primarily due to a lack of representation of these populations in genomics research'. The citation used for this claim is an oral abstract presented at an annual meeting which related to a small study examining patient
--	---

	and caregiver attitudes towards gene therapy for sickle cell disease. I am unclear as to why the authors are using this reference, namely it is about attitudes towards gene therapy, and it is solely focussed on sickle cell disease which is particularly common in people with an African or Caribbean background and this would surely provide evidence to the contrary of the authors assertion. This statement needs reviewed as does the reference. 4. Page 8, lines 167-169 states ‘reducing health inequalities may provide economic benefits for the National health Service (NHS), providing value for taxpayers’, with a references to a Kings Fund URL which no longer exists. I think this statement should be clearer to the reader – what are the authors implying that it will be better value for taxpayers and an economic benefit for the NHS? Is this through better engagement with employment, or through less cost to NHS? 5. Page 13, line 278 says that the co-production group will include those with ‘lived experience of the challenges being investigated’. If the challenges being investigated is the lack of inclusion of those from BAFDC in health and social care research, can the authors explain what constitutes as lived experience of this issue? 6. Page 18, line 406 details that a ‘data extraction form’ will be informed by the theoretical framework as a way of combing for evidence. I think it would be useful to see what this form looks like and the authors have not presented it – from the protocol I am left uncertain as to what information the authors will be extracting from relevant papers. Other 1. Page 5, Line 96-97. In the abstract, section ‘Strengths and Limitations’ it says ‘this study will help to inform the development of intervention reflected in the methods’, does this mean to say ‘interventions’? I would be happy to review any revisions of this manuscript.
--	--

REVIEWER	Raghavan, Raghu De Montfort University
REVIEW RETURNED	10-Jan-2024

GENERAL COMMENTS	This is an excellent realist review protocol for understanding the inclusion and participation of adults from Black African Diaspora Communities in health and social care research in the UK.
--

VERSION 1 – AUTHOR RESPONSE

Reviewer 1 comments	
Major comments	
1. My main concern with the current protocol is a lack of a clear search strategy, and no explicit references to the proposed dates of the review ie what timeline do the authors propose for this search. On Page 34 the reader is presented	A supplementary file (2) (see Line 417, paragraph 2, p 17) has been added and further detail describing the iterative search process explained in the manuscript (lines 429-432, para 1 , p 19). BAFDC is not a searchable term. It was an agreed term with the co-production group. Initial scoping and the pilot search

with 'search terms'. This is not a search strategy, and I believe usual convention would be to provide a clear example of the proposed search strategy that would be carried out, for example in Medline. I understand that the process of a realist review is iterative, however, an initial search will still be conducted and a protocol should detail that. Further, among the search terms, the terminology used by the authors themselves is not used, the BAFDC descriptor. I am also confused that the search terms include 'African American' when the review has a clear remit within the United Kingdom. Research which focuses on particular ethnic groups should be transparent with proposed terms that will be used in the search – the authors could inadvertently miss a particular population.	determined the most searchable terms in relation to inclusion and participation of Black communities in health and care research with an information specialist. The majority of available literature is from the US and so the term 'African American' was felt critical to incorporate, with the co-production group indicating that it would be important to include literature on this population due to a shared history as well as learning what may “work” there in relation to inclusion and participation. However, we have added the search terms: BAME, BME and Afro-Caribbean to widen the search, though this may require filtering out papers that are not specific in capturing evidence about Black African Diaspora Communities. Table 1. has been updated accordingly Though it is acknowledged that these terms are not representative, or popular, but may be found in some of the literature. As the review progresses, we will discuss the transferability of the US literature to the UK context as a research team and co-production group.
Page 22, line 513, states 'the review will be registered with PROSPERO' – this should be done before this article is considered for publication, so the reader has access and can review what has been registered by the authors.	The realist review protocol is now registered with PROSPERO . This has been added to Line 79, p 4 under the Ethics and Dissemination section Prospero registration number CRD42024517124 Line 526 , p23 , paragraph 2 amended to : The review is registered with PROSPERO.
Referencing needs reviewed. I believe the authors have referenced in Vancouver style, however from my reading, the references presented have not been numbered by the order in which they are cited and this is very confusing. For example on page 7, line 131 in the introduction there is references to [23,13, 27]. Yet this is the first time references 24 and 27 are presented, and therefore they should be 14 and 15. This needs corrected and the reference list updated consistently. The issue seems contained to the introduction section, but on correction this will change the entire ordering.	All references have been reviewed and updated to ensure they are in the correct order and referenced in Vancouver style according to the BMJ Open author guidance.

On page 18, lines 400-401 it states that ‘following a pilot search of the evidence it was felt that no time limit should be applied to the initial search’. If this is not a focused review on emerging evidence, why have the authors characterised their review as a ‘rapid’ review?	The time limit referred to in what was lines 400-401, para 1 , p 7 is in relation to the time limit applied to the search strategy due to a low return of papers from the pilot search. With this knowledge from the pilot search, it was felt critical to ensure coverage of literature beyond recent evidence, to capture the lack of progress and any key policy or strategies introduced (or not introduced). The previous response in Lines 403-405 now addresses this. We have discussed the characterisation of the review and would agree that it does not need to be classed as rapid. Therefore, the word rapid has been removed from the following lines: Title p1 Line 61,p3 Line 85, p4 Line 278, p12 Line 287, p12 Line 291-292 removed Line 520, para 2, p21 Lines 517-519, para 1, p22 have been revised as follows: Limitations may include not being able to access or include all sources identified in searches as well as a limit on the number of iterative searches conducted due to timescales of the research.
On pages 9 -10, the section ‘the science of inclusion’ is quite challenging to read/make sense of, and at times feels very unrelated to the topic of inclusivity in UK research. Authors may wish to reflect on whether this detailed analysis of US policy is needed.	This analysis of US documents and policies was felt to be important to understand “what works” , or has not “worked” elsewhere and builds onto the lack of progress in representation of Black people in health and care research. US policies have a long history of attempts at mandating inclusion (which have subsequently not improved representation) – these are important lessons to understand, despite the health care systems being different. In addition, the policies in the US were introduced in response to the abuse and experimentation endured by African Americans e.g. Tuskegee Syphilis study which has shaped attitudes and views towards health and care research in the UK, by Black communities. So, we conclude it is very important to include these details.
Minor comments	
Page 7, lines 138-140. In the introduction it states ‘Little progress has been made in improving health outcomes from individuals from BAFDC’. There is reference to MBBRACE-UK 2021 report which I believe is a great example of evidence for where little progress has been made to address ethnic	Line 138 : These references have been reviewed and re-numbered where out of sync. Updated references have been added to strengthen these points throughout Lines 138-149, para 2, p6 to para 1, p7 (refs 18-29) In reference to the NIHR references, these are used in the footnote . Ref 26 is the NIHR definition of participation with a reference to the NIHR glossary and definition of participation. Ref 27 is

differences in a particular health outcome. I am confused about the other references. The paper by Bello et al in Diabetologia, on quick review is a paper examining whether black West African men with type 2 diabetes present with greater hepatic and adipose tissue insulin sensitivity – for which there were no ethnic differences found. I do not understand how this provides evidence that there has been little progress made to improving health outcomes. I would recommend authors consider other evidence for poor outcomes among ethnic minority groups with respect to diabetic care, a good example on a quick review of the literature would be Mathur R, Farmer RE, Eastwood SV, Chaturvedi N, Douglas I, Smeeth L. Ethnic disparities in initiation and intensification of diabetes treatment in adults with type 2 diabetes in the UK, 1990–2017: A cohort study. PLoS medicine. 2020 May 15;17(5):e1003106. Further, references 19 and 20 are cited as evidence of limited progress, and these are links to the NIHR website. Reference 19 is to the NIHR glossary, under ‘p’– on review I could not identify any empirical evidence for poor health outcomes for ethnic minority groups/the BAFDC in this section of the NIHR website, and further I have no indication as to what the authors are referring to. Reference 20 directs me to guidance from the NIHR on public involvement in research. There is a multitude of excellent references which could be used to assert the claim made by the authors, but the ones selected are in my opinion very unclear. References I would recommend and I believe could improve this section would be: Devonport TJ, Ward G, Morrissey	an NIHR reference to the difference between participation and patient public involvement.
---	--

H, Burt C, Harris J, Burt S, Patel R, Manning R, Paredes R, Nicholls W. A systematic review of inequalities in the mental health experiences of Black African, Black Caribbean and black-mixed UK populations: implications for action. Journal of Racial and Ethnic Health Disparities. 2022 Jun 29;1-3.. Mukadam N, Marston L, Lewis G, Mathur R, Rait G, Livingston G. Incidence, age at diagnosis and survival with dementia across ethnic groups in England: A longitudinal study using electronic health records. Alzheimer's & Dementia. 2023 Apr;19(4):1300-7.. Mathur R, Dreyer G, Yaqoob MM, Hull SA. Ethnic differences in the progression of chronic kidney disease and risk of death in a UK diabetic population: an observational cohort study. BMJ open. 2018 Mar 1;8(3):e020145.	
On page 7, lines 141-142, it says 'black women are four times more likely to die during pregnancy...' and 'diabetes prevalence is up to three times greater in BAFDC' I assume this is in comparison to the white population in the UK, but I think this should be clarified for the reader.	Line 140 para 2, p6, revised to : In the UK, Black women are four times more likely to die during pregnancy, or up to one year after giving birth, than White women [23], type 2 diabetes prevalence
Page 8, Line 155-158 states 'it remains unclear as to how genetic determinants may contribute to the higher incidence of certain diseases amongst people from BAFDC, primarily due to a lack of representation of these populations in genomics research'. The citation used for this claim is an oral abstract presented at an annual meeting which related to a small study examining patient and caregiver attitudes towards gene therapy for sickle cell disease. I am unclear as to why the authors are using this reference, namely it is about attitudes towards gene therapy, and it is solely focussed	Now Lines 152-164 (p7), revised and improved references added : However, it remains unclear as to how genetic determinants may contribute to the higher incidence of certain diseases amongst people from BAFDC, primarily due to a lack of representation of these populations in genomics research which may have potentially serious consequences on treatment decisions [31,32]. For example, doses for the drug warfarin, a blood-thinning drug, have been reported as being prescribed based upon an individual's race and ethnicity, despite genetic studies mainly being conducted on European and White Americans [31]. There is also evidence to suggest that there is some misunderstanding amongst biomedical researchers around the differences between sociopolitical constructs like race and ethnicity and biological factors such as genetic ancestry, which can lead to estimates of genetic heredity being biased if environmental and social

on sickle cell disease which is particularly common in people with an African or Caribbean background and this would surely provide evidence to the contrary of the authors assertion. This statement needs reviewed as does the reference.	determinants of health are taken into consideration during analysis [33]. This statement has been strengthened with the following references : 31. Cerdeña, J., Grubbs V, . Non A, Racialising genetic risk: assumptions, realities, and recommendations. The Lancet, 2022. 400(10368): p. 2147-2154.doi: 10.1016/S0140-6736(22)02040-2 32. Fatumo, S Chikowore T, Choudhury A et al., et al., A roadmap to increase diversity in genomic studies. Nature Medicine, 2022. 28(2): p. 243-250.doi: 10.1038/s41591-021-01672-4 33. Phuong, J., Riches, N.O., Madlock-Brown, C., Duran, D., Calzoni, L., Espinoza, J.C., Datta, G., Kavuluru, R., Weiskopf, N.G., Ward-Caviness, C.K. and Lin, A.Y. (2022), Social Determinants of Health Factors for Gene–Environment COVID-19 Research: Challenges and Opportunities. Advanced Genetics, 3: 2100056. https://doi.org/10.1002/ggn2.20210005633.
Page 8, lines 167-169 states ‘reducing health inequalities may provide economic benefits for the National health Service (NHS), providing value for taxpayers’, with a references to a Kings Fund URL which no longer exists. I think this statement should be clearer to the reader – what are the authors implying that it will be better value for taxpayers and an economic benefit for the NHS? Is this through better engagement with employment, or through less cost to NHS?	We feel that this line is best removed without substantial evidence to indicate that this would be the outcome. Lines 170-173 removed
Page 13, line 278 says that the co-production group will include those with ‘lived experience of the challenges being investigated’. If the challenges being investigated is the lack of inclusion of those from BAFDC in health and social care research, can the authors explain what constitutes as lived experience of this issue?	Lines 279-283 para 1, p12 revised : It will be undertaken with a co-production group composed of people from BAFDC who have the lived experience of being Black as well as experience as: patients, members of the public, healthcare professionals, community/faith leaders, health and care researchers, research delivery staff, policymakers and funders of health and care research. Thus, they are aware of the challenges related to inclusion and participation in health and care research for people from BAFDC [62].
Page 18, line 406 details that a ‘data extraction form’ will be informed by the theoretical framework as a way of combing for evidence. I think it would be useful to see what this form looks like and the authors have not presented it –	As this is a protocol, the data extraction form will be published with the realist review. However, the following has been added to Lines 410-413, p17: The data extraction form will include standard information including: full reference, item type, country where

from the protocol I am left uncertain as to what information the authors will be extracting from relevant papers.	source was published, quality and the group being researched, though will be developed iteratively to meet the needs of the review. So, it is not possible to include a final draft here. The final form will be reported in the results paper.
Page 5, Line 96-97. In the abstract, section 'Strengths and Limitations' it says 'this study will help to inform the development of intervention reflected in the methods', does this mean to say 'interventions'?	Line 92 revised, p4 : "This study will help to inform the development of an intervention...."
Reviewer 2 comments	
This is an excellent realist review protocol for understanding the inclusion and participation of adults from Black African Diaspora Communities in health and social care research in the UK.	Thank you very much. We are delighted with your positive feedback.

VERSION 2 – REVIEW

REVIEWER	Bamford, Jordan The University of Manchester Division of Psychology and Mental Health
REVIEW RETURNED	12-Mar-2024
GENERAL COMMENTS	I commend the authors for their updated version of the paper. They have addressed each point raised and I recommend this paper is accepted.